# Functional Analysis of Membrane-Associated Scaffolding Tight Junction (TJ) Proteins in Tumorigenic Characteristics of B16-F10 Mouse Melanoma Cells

**DOI:** 10.3390/ijms25020833

**Published:** 2024-01-09

**Authors:** Eun-Ji Ko, Do-Ye Kim, Min-Hye Kim, Hyojin An, Jeongtae Kim, Jee-Yeong Jeong, Kyoung Seob Song, Hee-Jae Cha

**Affiliations:** 1Departments of Parasitology and Genetics, Kosin University College of Medicine, Busan 49241, Republic of Korea or ek3417@cumc.columbia.edu (E.-J.K.); dongkae97@gmail.com (D.-Y.K.); kmhmary93@naver.com (M.-H.K.); 322002@kosin.ac.kr (H.A.); 2Institute for Cancer Genetics, Columbia University Irving Medical Center, New York, NY 10032, USA; 3Department of Biomedical Sciences, Dong-A University, Busan 49315, Republic of Korea; 4Department of Anatomy, Kosin University College of Medicine, Busan 49241, Republic of Korea; kimjt78@kosin.ac.kr; 5Institute for Medical Science, Kosin University College of Medicine, Busan 49241, Republic of Korea; jyjeong@kosin.ac.kr (J.-Y.J.); kssong@kosin.ac.kr (K.S.S.); 6Department of Biochemistry, Kosin University College of Medicine, Busan 49241, Republic of Korea; 7Departments of Medical Life Science, Kosin University College of Medicine, Busan 49241, Republic of Korea

**Keywords:** tight junction (TJ) proteins, CRISPR/Cas9, cell–cell adhesion, tumorigenic characteristics

## Abstract

Tight junction (TJ) proteins (Tjps), Tjp1 and Tjp2, are tight junction-associated scaffold proteins that bind to the transmembrane proteins of tight junctions and the underlying cytoskeleton. In this study, we first analyzed the tumorigenic characteristics of B16-F10 melanoma cells, including cell proliferation, migration, invasion, metastatic potential, and the expression patterns of related proteins, after the CRISPR–Cas9-mediated knockout (KO) of *Tjp* genes. The proliferation of *Tjp1* and *Tjp2* KO cells significantly increased in vitro. Other tumorigenic characteristics, including migration and invasion, were significantly enhanced in *Tjp1* and *Tjp2* KO cells. Zonula occludens (ZO)-associated protein Claudin-1 (CLDN-1), which is a major component of tight junctions and functions in controlling cell-to-cell adhesion, was decreased in *Tjp* KO cells. Additionally, *Tjp* KO significantly stimulated tumor growth and metastasis in an in vivo mouse model. We performed a transcriptome analysis using next-generation sequencing (NGS) to elucidate the key genes involved in the mechanisms of action of *Tjp1* and *Tjp2*. Among the various genes affected by *Tjp* KO-, cell cycle-, cell migration-, angiogenesis-, and cell–cell adhesion-related genes were significantly altered. In particular, we found that the *Ninjurin-1* (*Ninj1*) and *Catenin alpha-1* (*Ctnna1*) genes, which are known to play fundamental roles in Tjps, were significantly downregulated in *Tjp* KO cells. In summary, tumorigenic characteristics, including cell proliferation, migration, invasion, tumor growth, and metastatic potential, were significantly increased in *Tjp1* and *Tjp2* KO cells, and the knockout of *Tjp* genes significantly affected the expression of related proteins.

## 1. Introduction

Tight junction proteins (Tjps), also known as zonula occludens (ZO), are scaffold proteins that are involved in the formation and maintenance of tight junctions and are critical for the integrity and barrier function of the epithelial and endothelial cell layers. Tjp1, Tjp2, and Tjp3 are peripheral membrane proteins that play crucial roles in linking the actomyosin cytoskeleton to other tight junction transmembrane proteins, such as occludins and claudins [1]. They also control apical–junctional organization and promote the formation of a functional barrier [2,3,4,5,6]. Previous studies on Tjp1 and Tjp2 have mainly focused on the regulation of epithelial barrier function. Tight junction protein compositions are involved in the maintenance of tight junction integrity, regulation of paracellular transport, and control of epithelial permeability. ZO proteins contain three N-terminal PDZ domains, followed by an SH3 domain, a GuK domain, and a carboxy-terminal end that includes both an acidic domain and a proline-rich region. PDZ domains are abundant protein interaction modules that frequently recognize short amino acid motifs at the C-termini of target proteins. They play a regulatory role in various biological processes, including transport, ion channel signaling, and other signal transduction systems [7,8].

Tjp1 and Tjp2 also interact with cytoplasmic proteins to form higher-order molecular structures at junctions, where various proteins involved in signaling and transcriptional modulation are recruited [9,10,11]. These proteins also include those involved in epithelial proliferation and differentiation, and tight junction protein compositions are known to function beyond regulating cell permeability [12]. Moreover, Tjp1 and Tjp2 have been reported to be associated with signaling pathways and cytoskeletal organization. Tjp1 interacts with various signaling molecules and cytoskeletal components, thereby participating in cell signaling and cytoskeletal organization. A number of functional studies have investigated the role of Tjp1 in cell polarity, migration, and cytoskeletal dynamics. Previous studies have reported the involvement of Tjp1 in cadherin-based cell adhesion via direct binding to catenin and actin filaments in exogenous E-cadherin (EL cells) [13]. Depletion of Tjp1 results in a decrease in non-muscle myosin-2B (NM2B) integration at the junction [14,15,16], an increase in conical stiffness [17], a loss of the twisting of the tight junction membrane [14,18], and an increase in joint contractility, combined with tissue changes in the conical actin filament [19,20]. Tjp1 partly influences cell proliferation by binding transcription factors that can localize to both adhesion complexes and the nucleus, where they play a role in regulating gene expression [21,22,23]. In a recent study on the regulation of the mechanical properties of apical and junctional membranes, the organization of apical actomyosin filaments and the junctional assembly of Tjp1 were reported to undergo changes and interactions in barrier structures in CRISPR-knockout (KO) epithelial cell lines [24]. Furthermore, Tjp1 and Tjp2 are reported to be associated with several diseases, including cancer, autoimmune disorders, and gastrointestinal disorders. Celiac disease (CD) is a representative autoimmune and inflammatory disease in which tight junctions open in the early stages, leading to severe intestinal damage [25,26,27,28]. Recently, the association between inflammation and TJ proteins has been studied. In particular, it has been reported that ZO-1 expression is reduced in inflammation-related human lung disease [8]. Type 1 diabetes is also associated with tight junctions. Observations in animal models have suggested that increased infiltration occurs prior to the appearance of histological or obvious signs of diabetes. Previous studies support these findings by reporting that zone-dependent intestinal permeability increases in the same rat model two to three weeks before the onset of type 1 diabetes [29].

The effect of Tjp on the tumorigenic characteristics of cancer cells varies depending on the specific context and type of cancer. In some studies, the loss or downregulation of Tjp1 has been associated with increased tumorigenic characteristics and cancer progression. A reduced expression of Tjp1 has been observed in various cancer types, including breast, ovarian, and colorectal cancers, and the loss of Tjp1 may contribute to the disruption of cell–cell adhesion and increase cell migration, invasion, and metastasis, all of which are hallmarks of aggressive cancer behavior [30,31,32]. However, there are also studies suggesting that Tjp1 has tumor suppressive properties. Tjp1 expression is capable of inhibiting cell proliferation, inducing cell cycle arrest, and promoting apoptosis in cancer cells. In some instances, Tjp1 has been found to regulate key signaling pathways involved in tumor growth and survival, such as the Akt and ERK pathways [33,34]. Therefore, although it is difficult to make a generalized statement, in certain cancer types, the knockout of Tjp1 may indeed lead to an increase in tumorigenic characteristics and cancer progression. However, in other contexts, Tjp1 loss may contribute to a decrease in tumorigenic characteristics. In humans, Tjp1 and Tjp2 are known to play a role in the invasion and metastasis of cancer by regulating the expression of Tjp1 and Tjp2 and causing changes in barrier function due to structural alterations in tight junctions [35]. Early studies found a correlation between reduced tight junctions and tumor differentiation [36,37,38], and research on the functional roles of tight junctions in tumor development has increased, although most studies have been conducted in vitro [30,39]. In breast cancer, Tjp1 is reduced in poorly differentiated tumors and is associated with an increase in tumor grade and TNM (tumor-nodal) status [32]. Studies exploring the effects of tight junction molecules on tumor progression are currently accumulating [35,40]. Although TJ proteins are closely related to cancer and other diseases, studies on their mechanisms of action are scarce. A recent study revealed that the expression of Tjp1 is decreased in human lung diseases associated with inflammation [8] and in pancreatic cancer cell lines, and the knockdown (KD) of Tjp1 and claudin-1 contributes to tumor migration and invasion in xenograft tumors [41].

A tight junction protein is a protein that plays a role in cell-to-cell binding. However, the investigation into how this binding influences the characteristics of cancer growth, metastasis, migration, invasion in cancer cells has not yet been conclusively conducted.

In this study, we first knocked out *Tjp1* and *Tjp2* in B16-F10 mouse melanoma cells using the CRISPR–Cas9 system and analyzed the tumorigenic characteristics, such as invasion and migration, in vitro. Additionally, we analyzed the effects of *Tjp1* and *Tjp2* KO on tumor growth and experimental metastasis in an in vivo model. We confirmed TJ assembly by re-expressing *Tjp* in KO cells. An RNA-seq analysis was used to identify genes that are modulated by TJ proteins and are associated with tumorigenic characteristics and adhesion.

## 2. Results

### 2.1. Knockout of Tjp1 and Tjp2 in Mouse Melanoma Cell Lines

To identify the role of Tjp1 and Tjp2 in the tumorigenic characteristics of mouse melanoma, *Tjp1* and *Tjp2* KO B16-F10 melanoma cell lines were generated using the CRISPR/Cas9 gene editing system. gRNA was designed with the *Tjp1* and *Tjp2* exon 5 region, which encodes the most functional TJ protein, and *Tjp1* and *Tjp2* KO stable clones were selected through hygromycin selection. The locations and sequences of *Tjp*1 and *Tjp*2 gRNAs are displayed in Figure 1A and Appendix A. After the selection of CRISPR/Cas9 system-transfected cells, *Tjp1* and *Tjp2* KO clones were confirmed with RT-PCR (Appendix A). The RNA levels of *Tjp1* and *Tjp2* were confirmed using real-time PCR, and *Tjp1* and *Tjp2* expression was significantly reduced in each KO clone (Figure 1B). Genomic PCR (Figure 1C) containing the gRNA-selected region with specific primers (Appendix A) confirmed the complete removal of the TJ protein genes, while other derivatives at different loci remained unaffected. The protein levels of Tjp1 and Tjp2 were also significantly reduced in KO cells (Figure 1D). To further investigate the impact and mechanism of *Tjp* gene KO, we generated re-expressing cells by overexpressing *Tjp1* and *Tjp2* in each KO cell line. The expression of *Tjp1* and *Tjp2* at the RNA level was partially recovered in each re-expressed cell line (Figure 1E), and IF and Western blot analyses confirmed that the protein levels were also recovered in each re-expressed cell line (Figure 1F,G). The expression of claudin-1, which interacts with Tjps to form tight junctions, was significantly reduced in both *Tjp1* and *Tjp2* KO cells; however, the reduced expression level was not recovered after re-expression in both *Tjp1* and *Tjp2* KO cells (Figure 1G).

### 2.2. Decreased Expression of Tight Junction Proteins Stimulates Tumorigenic Characteristics, Including Invasion, Migration, Cell Proliferation, Tumor Growth, and Experimental Metastasis

We conducted in vitro migration and invasion assays to examine the effects of *Tjp* KO on the migration and invasion of B16-F10 melanoma cells. As shown in Figure 2A, the migration and invasion of *Tjp* KO cells was significantly increased compared to those of the MOCK cells. These results suggest that the depletion of *Tjp1* and *Tjp2* stimulates the invasion and migration of B16-F10 melanoma cells. A significant increase in cell proliferation was the most prominent characteristic of *Tjp* KO cells. As shown in Figure 2B, cell growth was significantly higher in *Tjp1* and *Tjp2* KO cells, whereas cell growth was recovered in *Tjp1* and *Tjp2* re-expressed cells compared with MOCK cells. 

We examined the effects of *Tjp1* and *Tjp2* KO expression on tumor growth and metastasis by subcutaneously and intravenously injecting separate groups of mice (10 per test group) with MOCK, *Tjp1* and *Tjp2* KO, or *Tjp1* and *Tjp2* re-expressed B16-F10 melanoma cells. Fourteen days after the subcutaneous injection, the mean tumor size of MOCK B16-F10 cells was 11.32 mm (95% CI_6.72 to 15.57). The mean tumor size was 16.1 mm (95% CI_ 13.84 to 20.33 mm) for mice injected with *Tjp1* KO B16-F10 cells and 14.9 mm (95% CI_ 9.71 to 18.56 mm) for mice injected with *Tjp1* re-expressed B16-F10 cells. Furthermore, the mean tumor size was 14.5 mm (95% CI_ 9.37 to 20.6 mm) for mice injected with *Tjp2* KO B16-F10 cells and 12.8 mm (95% CI_ 9.94 to 20.23 mm) for mice injected with *Tjp2* re-expressed B16-F10 cells. After comparing the mean tumor sizes of all treatment groups, we discovered that tumor growth was significantly increased in the *Tjp1* and *Tjp2* KO injected groups and growth rates recovered in the *Tjp1* and *Tjp2* re-expressing groups. These results confirmed that tumor growth was significantly promoted in *Tjp1* and *Tjp2* KO mice in vivo (Figure 2C). We further investigated whether *Tjp1* and *Tjp2* KO stimulated the metastatic potential of B16-F10 melanoma cells. Tumor cells were injected into the tail vein of the mice to evaluate their differential colonization abilities in the lungs. Three weeks after the mice were intravenously injected with the cells, the mean number of metastatic lung nodules was visually monitored (Figure 2D). Compared with the MOCK mice, significant increases in the size and number of lung nodules were observed in *Tjp1* and *Tjp2* KO-injected mice, but metastasis rates recovered in re-expressing cells. These results suggest that *Tjp1* and *Tjp2* KO not only promotes tumor growth but also enhances the metastatic potential of B16-F10 melanoma cells.

### 2.3. RNA Sequencing (RNA-Seq) for Analysis of Gene Expression Profile

The cluster analysis of mRNAs related to *Tjp* KO is displayed in Figure 3. To identify the functional mechanism of TJ proteins in tumorigenic characteristics, the transcriptomes of *Tjp1* KO and *Tjp2* KO B16-F10 melanoma cells were analyzed and compared with those of MOCK cells using RNA sequencing. Heat maps of gene expression changes are displayed in Figure 3A for the *Tjp1* KO group (89 downregulated and 55 upregulated; top panel) and in Figure 3B for the *Tjp2* KO group (90 downregulated and 54 upregulated; top panel). The expression fold changes (FC) and various category-specific analyses of *Tjp1* and *Tjp2* KO cells were compared with those of MOCK cells using the ExDEGA program (Figure 3C,D). Among these categories, we analyzed the genes associated with tumor characteristics and the cell barrier. To identify the meaningful targets of tight junction proteins, Venn diagrams were analyzed in *Tjp1* and *Tjp2* KO B16-F10 melanoma cells. Approximately 49,196 transcripts were identified, and 144 of these differentially expressed genes (DEGs) were significantly different (Figure 3E).

Among the 144 DEGs, genes related to cell cycle, cell migration, angiogenesis, and cell–cell adhesion appeared to be related to the tumorigenic characteristics of tight junction proteins. From the multitude of genes expressed in the context of *Tjp1* and *Tjp2* KO, we have chosen genes that commonly exhibit increases and decreases in tumorigenesis-related characteristics, including cell cycle, cell migration, angiogenesis, and cell–cell adhesion. This selection is based on consideration of cancer-related phenotypes that manifest (Table 1 and Table 2). The genes related to *Tjp1* and *Tjp2* KO, *Ninj1*, and *Ctnna1*, which are important cell–cell adherence factors, were significantly downregulated in both *Tjp* KO groups. As shown in Figure 4A,B, *Ninj1* and *Ctnna1* were downregulated at the mRNA level. IF confirmed that Ninj1 and Ctnna1 were downregulated at the protein level (Figure 4C).

## 3. Discussion

The role of tight junction protein in cancer has been extensively studied; however, its significance remains controversial. Tight junction proteins contribute to cell proliferation via several mechanisms. In contrast to our observations, the TJP-1 protein was found to be highly expressed in adenocarcinoma samples compared to healthy tissues [42]. Importantly, high claudin expression in human cancer tissues is inconsistent with boundary localization and barrier regulation [43]. In addition, previous studies distinguish between claudin expression level and tight junction formation and conclude that deleting ZO proteins results in no tight junctions despite normal claudin levels in mouse mammary gland tumor Eph4 cells [44]. In our results, the expression of claudin was significantly reduced upon Tjp KO cells. However, the re-expression of Tjp did not show a significant expression of claudin difference compared to that of Tjp KO cells. It is worth noting that B16F10 cells, being highly aggressive melanoma cells, exhibit variations that may be attributed to their cell type.

Overall, previous studies have suggested that ZO proteins contribute to the control of the contact regulation of cell proliferation. Decreased TJP-1 expression is correlated with increased invasiveness in breast cancer [45], colorectal cancer [46], and cancers of the human digestive tract [47]. Additionally, TJP-1 has been reported to be involved in epithelial–mesenchymal transition (EMT) processes associated with tumor invasion [48]. Therefore, TJP-1 is believed to play a significant role in the processes underlying tumor growth, and its expression is closely associated with patient prognosis. TJP-2 is also a tumor suppressor protein. Previous studies have reported that TJP-2 exhibits a loss of localization at the cell borders and displays cytoplasmic staining in testicular carcinoma in situ and in bronchopulmonary cancers. This abnormal localization of TJP-2 is associated with the disruption of the blood–testis barrier [49] and the invasive characteristics of lung cancer cells in vitro [50]. The expression of TJP-2 has also been found to be downregulated in various carcinomas, such as breast [51,52] and pancreatic [53] as well as in hypoxia-resistant cancer cell lines derived from scirrhous gastric carcinoma [54]. Notably, patients with incompletely enhanced glioblastoma multiforme (GBM), who exhibit an increased survival rate, display higher levels of TJP-2 expression than those with completely enhanced GBM, which is associated with shortened survival [55]. In the context of tumorigenesis, the altered expression or dysfunction of tight junction proteins has been implicated in increased tumor invasiveness and metastasis. The loss of tight junction integrity promotes the migration and invasion of cancer cells through tissues. Furthermore, the disruption of tight junctions can lead to changes in cellular polarity, aberrant signaling pathways, and enhanced angiogenesis, all of which are hallmark features associated with tumor progression [12]. However, in Madin–Darby Canine Kidney (MDCK) cells, TJP-1 KO or double knockdown (dKD) decreased the migration of individual cells within a confluent monolayer. In addition, the velocity of single TJP-1 and TJP-2 dKD cells in the absence of cell–cell contact was even higher than that of single WT cells [56]. These results suggest that the newly discovered function of TJP1 and TJP2 also plays a role in efficient population cell migration by maintaining tissue fluidity and regulating proliferation. In this study, we successfully generated *Tjp1* and *Tjp2* KO melanoma cells for the first time using the CRISPR/Cas9 system and demonstrated that TJ proteins are not only important for barrier function but also have a significant effect on cancer progression. We observed significant increases in cell migration, invasion, and proliferation in vitro. In vivo experiments revealed a significant increase in both the size and metastasis of cancer cells compared to the MOCK groups. Our results confirmed that *Tjp* KO in B16-F10 cells increased cancer progression and improved tumor characteristics (Figure 5). These results are equivalent to those of previous reports that suggested that the tight junction protein, zonula occludens (ZO)-1, regulates cell proliferation and gene expression [57]. Moreover, studies on metastasis and tight junction proteins have demonstrated that the expression of TJP1 mRNA is correlated with lymph node metastasis in patients with human bladder cancer [58]. Tjp2 may represent a novel target for molecular therapies aimed at preventing the invasion and metastasis of hamster pancreatic cancer [59]. Tight junction proteins could have a fundamental role in preventing the metastasis of breast cancer cells [35]. In this study, we explored the post-extravasation process through experimental metastasis. Even in the step where the binding force is weakened, allowing for free cells to penetrate blood vessels, they can easily exit the host cells without entrapment, in contrast to general cancer cells. There is a high likelihood of avoiding the binding of immune cells, and metastasis may be promoted by the tight junction proteins’ knockout even during growth in secondary tumors.

We also found that *Ninj1* and *Ctnna1*, which are important factors for cell–cell adhesion in a transcriptome analysis, were significantly reduced in the Tjp KO group. Ninj1 is known to play an important role, not only in cell–cell adhesion but also in immune function. Ninj1 also plays a crucial role in pulmonary fibrosis by promoting the interactions between macrophages and alveolar epithelial cells [60]. Ninj1 expression leads to macrophage activation in intestinal inflammatory conditions [61] and its upregulation in macrophages enhances cell–cell and cell–matrix adhesion in the hyaloid vascular system [62]. Its role in immunity and inflammatory reactions potentially affects the expression of TJ proteins. Ctnna1 (catenin) is primarily associated with adherens junctions, which are cell–cell junctions that link neighboring cells. It is part of a complex of proteins that helps to anchor and connect cells to one another. This complex, known as the adherens junction, plays fundamental roles in cell adhesion, signaling, and tissue integrity. It forms a complex with other proteins, including cadherins and catenin, and connects its cytoplasmic domain to the actin cytoskeleton. This linkage is essential for the stability and integrity of adherens junctions [63,64,65]. In particular, TJP-1 is an actin filament (F-actin)-binding protein located at a close junction that connects claudin to the actin cytoskeleton of epithelial cells. In non-cortical cells without tight junctions, TJP-1 is localized to adhesive junctions (AJ) and can indirectly link cytosines to the actin cytoskeleton via beta- and alpha-catenin [66]. Although a direct relationship between Tjp2 and Ctnna1 has not yet been reported, our study revealed a reduction in Ctnna1 expression in both Tjp1 KO and Tjp2 KO cells. Ctnna1 and tight junction proteins are distinct but complementary components of cell junctions that contribute to the overall stability, integrity, and functionality of tissues and cell layers in the body. However, the relationships between Ninj1, Ctnna1, and Tjp remain unclear. Further research is needed to determine how *Tjp1* and *Tjp2* KO regulates the protein expression of Ninj1 and Ctnna1 and how the functions of Tjp and these proteins are correlated.

Our study identifies genes modified by KO and highlights the genetic characteristics of tumors. Specifically, Tjp1 and Tjp2 play crucial roles in accurately regulating these processes. Previous studies have reported that the inhibition of ZO-1 restrains the proliferation and invasion of oral squamous carcinoma cells [67] while enhancing the cell proliferation and invasion capacity of endometrial cancer [68], pancreatic cancer [41], and liver cancer [69]. Recently, a known regulatory mechanism has emerged in pancreatic cancer. Mechanistically, the zinc transporter protein 4 (ZIP4) exhibits high expression, suppressing ZO-1 and claudin-1 expressions by modulating the mesenchymal cell marker zinc finger E-box binding homeobox 1 (ZEB1). ZEB1 directly binds to the promoters of ZO-1 and claudin-1, repressing their transcription. Inhibition of ZIP4 elevates the expression levels of ZO-1 and claudin-1, as well as the phosphorylation levels of focal adhesion kinase (FAK) and Paxillin—two molecules related to cell adhesion and motility. The subsequent silencing of ZO-1 or claudin-1 rescues the phosphorylation levels and mitigates the phenomena of attenuated cell proliferation and invasion. The controversies surrounding the regulation of cancer cell biological processes by ZO proteins may be explained through the complicated and heterogeneous signal-regulating backgrounds of different cancer cells [70].

In addition, various studies have revealed binding genes in the context of previous melanoma treatments. Tight junction proteins regulate the paracellular transport of molecules, but the staining of a tissue microarray revealed that claudin-1 was overexpressed in melanoma and aberrantly expressed in the cytoplasm of malignant cells, suggesting a role other than transport [71]. The normal phenotype and controlled proliferation of melanocytes is strictly regulated by keratinocytes via E-cadherin. The extracellular domains of separate E-cadherin molecules are tethered together, and the intracellular domain is anchored to the actin fibers of the cytoskeleton via a complex of catenins [72]. The barrier molecule junctions plakoglobin, filaggrin, and dystonin play roles in melanoma growth and angiogenesis [73]. Further exploration is necessary to assess the potential translational implications of these findings in clinical melanoma treatment.

These other regulatory genes are involved; we identified expression changes in several genes involved in the cell cycle, cell migration, angiogenesis, and cell–cell adhesion in *Tjp1* and *Tjp2* KO cells. Based on these results, a further correlation analysis with genes whose functions have not been extensively studied is also needed. In addition, another characteristic feature indicated by our results is the promotion of cell division and metastasis in *Tjp1* and *Tjp2* KO regarding the association between tight junction proteins and cell division; these studies remain controversial, lacking clear evidence. Therefore, further follow-up studies are warranted.

## 4. Materials and Methods

### 4.1. Generation of Knockout Cell Line with CRISPR–Cas9

Guide RNA (gRNA) sequences for the CRISPR/Cas9 system were designed using the CRISPR design provided by Bioneer (Daejeon, Republic of Korea). The insert oligonucleotides for tight junction protein-1 gRNA #1 were 5′-ACGGACCGCCTGTCCGAGCGAGG-3′ and 5′-GAAGCTTATGAACCCGACTACGG-3’ for tight junction protein-2 gRNA #2. The complementary oligonucleotides for gRNAs were annealed and cloned into the CRISPR/Cas9-Puro and pRGEN-Cas9-CMV/T7-Hygro-EGFP vectors (Toolgen, Seoul, Republic of Korea). Mouse melanoma B16-F10 cells were transfected with CRISPR/Cas9+gRNA #1 and #2. Approximately 18 h after transfection, the cells were treated with 100 μg/mL hygromycin for 2 days. After 2 weeks, colonies were isolated using cloning cylinders, RT-PCR, qRT-PCR, and Western blotting.

### 4.2. Cell Culture

Mouse melanoma B16-F10 cells were obtained from American Type Culture Collection (ATCC; Manassas, VA, USA). B16-F10 cells were cultured in RPMI1620 containing 10% fetal bovine serum (FBS) (Invitrogen, Carlsbad, CA, USA), 1% penicillin, and L-glutamine (Gibco, Brooklyn, NY, USA). All cell lines were maintained at 37 °C in a humidified atmosphere containing 5% CO_2_ and 90% humidity.

### 4.3. Plasmid Construct for Over-Expression

The plasmid construct used in the present study was the pcDNA3.1+/C-(K)-DYK expression vector. Tjp1_OMu20010 (catalogue number U9042HE240-1) and Tjp2_OMu15570 (Cat. NO NM_001198985.2) were purchased from Genescript (Piscataway, NJ, USA). Plasmid transfection was carried out using the Lipofectamine 2000 reagent (Thermo Fisher Scientific, Waltham, MA, USA) and according to the Lipofectamine 2000 protocol.

### 4.4. RT-PCR, qRT-PCR, and Genomic PCR (Polymerase Chain Reaction)

RNA was extracted from the cells using TRIzol Reagent (Invitrogen), cDNA was synthesized using a cDNA synthesis kit (Bioneer, Daejeon, Republic of Korea), qRT-PCR was performed using TB Green Premix Taq (Takara, Japan), and the results were recorded using QuantStudio 3 (Thermo Fisher Scientific). The relative gene expression levels were quantified based on the 2^−ΔΔCt^ method and normalized to the reference gene, *GAPDH*. The primer sequences for *Tjp* genes are as follows: *Tjp1* sense, 5′-GCC TCT GCA GTT AAG CAT-3′; antisense, 5′-AAG AGC TGG CTG TTT TA-3′; *Tjp2* sense, 5′-ATG GGA GCA GTA CAC CGT GA-3′; antisense, 5′-TGA CCA CCC TGT CAT TTT CTT G-3. mGAPDH was used as a control (sense primer, 5′-CTCATGACCACAGTCCAT-3′; antisense primer, 5′-CACATTGGGGGTAGGAAC-3′). The *Tjp1* RT-PCR cycling conditions were as follows: 94 °C for 2 min to activate DNA polymerase; 38 cycles at 94 °C for 1 min; 58 °C for 1 min; 72 °C for 1 min; and 72 °C for 10 min for post-elongation. The *Tjp2* RT-PCR cycling conditions were as follows: 94 °C for 2 min to activate DNA polymerase; 38 cycles of 94 °C for 1 min; 60 °C for 1 min; 72 °C for 1 min; and 72°C for 10 min for post-elongation. Genomic PCR was performed using chromosomal DNA from the cells and PCR primers. *Tjp1* sense, 5′-TGT GTT GGG GAA ATG TGC TG-3′, antisense 5′-CTG GCC CAA CAT TTC TTG CT-3′; *Tjp2* sense, 5′-ACG ACC GAG GTT TTG AAG TG-3′, antisense 5′-TGT TGG CCC TTG TGT TCA TG-3′. The *Tjp1* and *Tjp2* genomic PCR cycling conditions were as follows: 94 °C for 10 min to activate DNA polymerase; 30 cycles of 94 °C for 1 min; 60 °C for 1 min; 72 °C for 1 min; and 72 °C for 5 min for post-elongation. The products were analyzed on a 1.8% agarose gel and photographed under LED light.

### 4.5. Western Blot Analysis

Western blotting was performed as previously described [74]. The following antibodies were used: Tjp1 (1:1000, Invitrogen, USA); ZO-2 (1:1000, Cell Signaling, Denver, MA, USA); Claudin-1 (1:1000, Invitrogen, USA); and GAPDH (1:5000, R&D Systems, Minneapolis, MN, USA). Secondary antibodies linked to HRP (RSA1122, RSA1221, BioActs, Incheon, Republic of Korea) were also used.

### 4.6. In Vitro Migration and Invasion Assays

In vitro migration and invasion assays were performed as previously described [75]. Briefly, transwell chambers containing membranes with an 8 μm pore size (Invitrogen) were used for both assays. For the migration assay, 600 μL of conditioned medium, which was obtained by culturing B16-F10 cells for 18 h in serum-free RPMI, was placed in the lower chambers of each well. B16-F10 cells (1 × 10^4^ cells) were resuspended in 100 μL of serum-free RPMI and placed in the upper chambers of each well. The chambers were incubated for 18 h at 37 °C, and the cells in the lower chambers were fixed and stained with Diff Quit (Sysmex, Tokyo, Japan) following the Diff Quit protocol. The invasion assay was performed in a similar fashion, except that the upper surface of the transwell filter was coated with 20 µL of 0.5 mg/mL Matrigel (BD Biosciences, Bedford, MA, USA) before the cells were added to the upper chambers. All experiments were repeated at least three times, and each data point was measured in triplicate. Mean values and 95% confidence intervals (CIs) were calculated.

### 4.7. Cell Proliferation Assay

The in vitro cell proliferation assay was performed as described previously [76]. Briefly, cells (1 × 10^4^ cells per well) were plated in complete medium in a 6-well plate and incubated for 72 h. The cells were then harvested, and cell proliferation rates were measured by counting viable cells using the trypan blue dye exclusion method.

### 4.8. Immunofluorescence (IF)

Immunofluorescence analysis was performed as described previously [77]. The chamber slides (1 × 10^4^ cells) were incubated for 48 h at 37 °C, and cells were fixed with methanol. Slides were then permeabilized by incubation in 10% normal serum in phosphate buffered saline (PBS) for 1 h to block nonspecific antibodies. Slides were stained with antibodies to ZO-1 (1:500 dilution, Invitrogen, USA), ZO-2 (1:500 dilution, Cell Signaling, USA), Ninj1 (1:100 dilution, ABclonal, Woburn, MA, USA) [78], and Ctnna1 (1:500 dilution, Cell Signaling, USA) overnight at 4 °C. After washing thrice with PBS for 5 min, the slides were incubated with secondary antibodies, including Alexa Fluor 546 anti-mouse and Alexa Fluor 488 anti-rabbit antibodies (Invitrogen, Carlsbad, CA, USA). Specimen epifluorescence was determined using a confocal laser scanning microscope (LSM510 META; ZEISS, Jena, Germany). Confocal images were analyzed using AlphaEase FC image analysis IS-2200 software (Alpga Innotecgh, Randburg, Gauteng, South Africa). Fluorescence was measured using a Leica DMi8 microscope (Leica, Wetzlar, Germany).

### 4.9. Subcutaneous Tumor Growth and Experimental Metastasis Assays

Six-week-old female C57BL/6 mice (NCI, Frederick, MD, USA) were used for the tumor xenograft assay. Cells (2 × 10^5^ cells/200 μL; 10 mice per group) were injected subcutaneously, and tumor diameters were measured every 2 days for 2 weeks post-injection using digital calipers. The metastatic potential of *Tjp1*, *Tjp2* KO, and re-expressing cells was also examined using a lung colonization assay as previously described [75]. Briefly, C57BL/6 mice (10 mice per group) were infected with either the control or KO and re-expressing cells (1 × 10^5^ cells/200 μL) via the tail veins. Three weeks later, the mice were sacrificed via asphyxiation with CO_2_, and their lungs were removed. The metastatic nodules on the lung surfaces were counted. Animal experiments were approved and conducted under the guidance of Kosin University College of Medicine Institutional Animal Care and Use Committee (KUCMIACUC; KMAP-22-12) [79].

### 4.10. Library Preparation and Sequencing

Libraries were prepared from total RNA using the NEBNext Ultra II Directional RNA-Seq Kit (NEW ENGLAND BioLabs, Inc., Hitchin, UK). mRNA was isolated using a Poly(A) RNA Selection Kit (LEXOGEN, Inc., Wien, Austria). The isolated mRNAs were used for cDNA synthesis and shearing following the manufacturer’s instructions. Indexing was performed using Illumina indexes 1–12. Enrichment was performed using PCR. Subsequently, libraries were examined using a TapeStation HS D1000 Screen Tape (Agilent Technologies, Amstelveen, The Netherlands) to evaluate the mean fragment size. Quantification was performed using a library quantification kit and StepOne Real-Time PCR System (Life Technologies, Inc., Carlsbad, CA, USA). High-throughput sequencing was performed via paired-end 100 sequencing using a NovaSeq 6000 (Illumina, Inc., San Diego, CA, USA).

### 4.11. Data Analysis

Transcriptome analysis was performed using Ebiogen (Ebiogen Inc., Seoul, Republic of Korea). Quality control of the raw sequencing data was performed using Fast QC. Adapter and low-quality reads (<Q20) were removed using FASTX_Trimmer and BBMap. Trimmed reads were mapped to the reference genome using TopHat. The Read Count (RC) data were processed based on the fragments per kilobase per million (FPKM+G) + geometric normalization method using EdgeR within R. Fragments per kilobase per million (FPKM) reads were estimated using Cufflinks. Data mining and graphic visualization were performed using ExDEGA v4.1 (Ebiogen Inc., Seoul, Republic of Korea).

### 4.12. Statistical Analysis

Descriptive statistics were calculated for the patient characteristics. Statistical significance was defined as a two-tailed *p*-value < 0.05. The fluorescence intensity was read using image analysis software, and the intensity was measured to calculate the mean values and 95% CIs. The statistical significance of the differences among groups was determined using a two-tailed Student’s *t*-test.

## 5. Conclusions

In conclusion, *Tjp1* and *Tjp2* play a significant role in cell proliferation, migration, cancer growth, and metastasis, involving various genes.

## Figures and Tables

**Figure 1 ijms-25-00833-f001:**
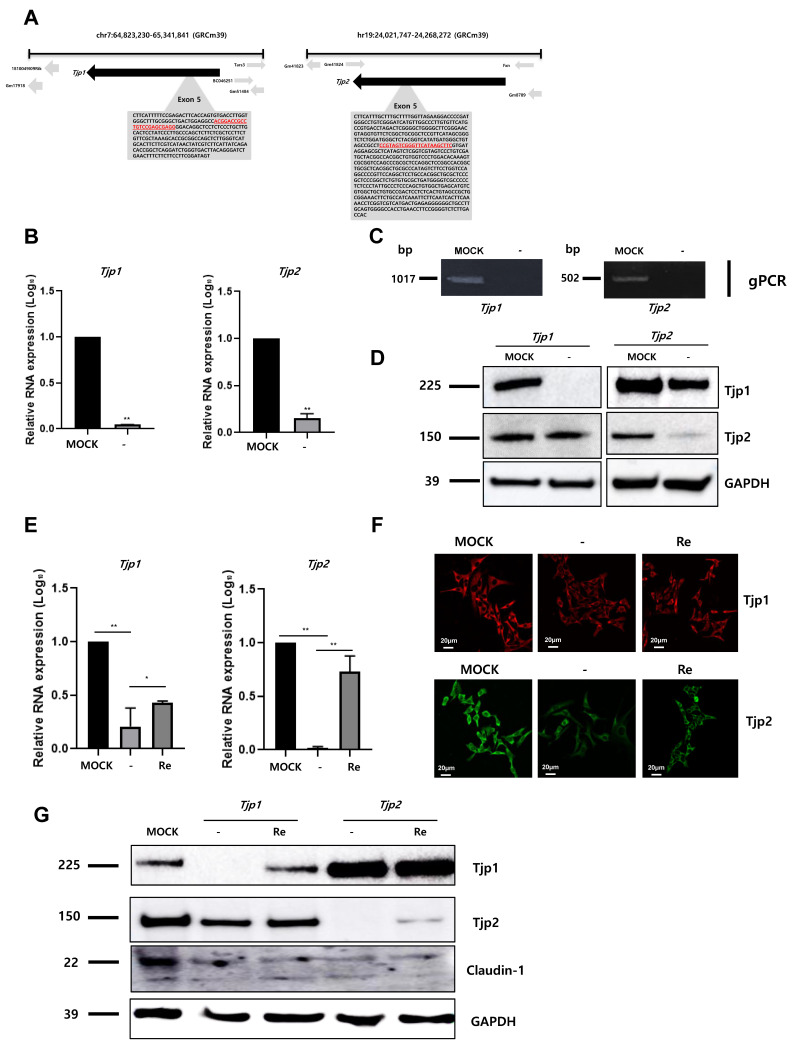
Knockout of tight junction (TJ) protein expression in B16-F10 melanoma cells. (**A**) Selection of guide RNAs (gRNAs, in red) for the targeting of *Tjp1* and *Tjp2* in knockout system. gRNA was designed with the *Tjp*1 and *Tjp*2 exon 5 region, which encodes the most functional TJ proteins. (**B**) Expressions of *Tjp*1 and *Tjp*2 RNA in *Tjp*1 and *Tjp*2 knockout (KO) cells, respectively. Real-time polymerase chain reaction (RT-qPCR) performed for general region of *Tjp* genes. Real-time PCR reactions were coupled to melting-curve analysis to confirm the amplification specificity. Non-template controls were included for each primer pair to check for any significant level of contaminants. Real-time PCR was performed in three independent experiments with three different samples per group. The mRNA expression levels were calculated by normalizing to GAPDH and presented relative to the control with calculated mean values and 95% confidence intervals. The statistical significance of differences between groups was determined using a two-tailed Student’s *t*-test. ** *p* < 0.01. (**C**) Genomic PCR was performed for specific regions of *Tjp* derivatives. (**D**) Protein expression of Tjp1 and Tjp2 in *Tjp1* KO and *Tjp2* KO B16-F10 cells. Western blotting was performed to analyze the protein levels of TJ proteins. (**E**) Relative mRNA expression levels of *Tjp*1 KO and *Tjp*2 KO and re-expressed B16-F10 melanoma cells. The mRNA expression levels were determined with real-time qPCR and calculated by normalizing to GAPDH. * *p* < 0.05, ** *p* < 0.01. (**F**) Immunofluorescence (IF) staining of the tight junction protein expressions in *Tjp1* and *Tjp2* KO cells and re-expressed cells. (**G**) Protein expressions of Tjp1, Tjp2, and Claudin-1 in MOCK, *Tjp*1KO, *Tjp*2 KO, and re-expressed B16-F10 melanoma cells. Protein expressions were analyzed with Western blot.

**Figure 2 ijms-25-00833-f002:**
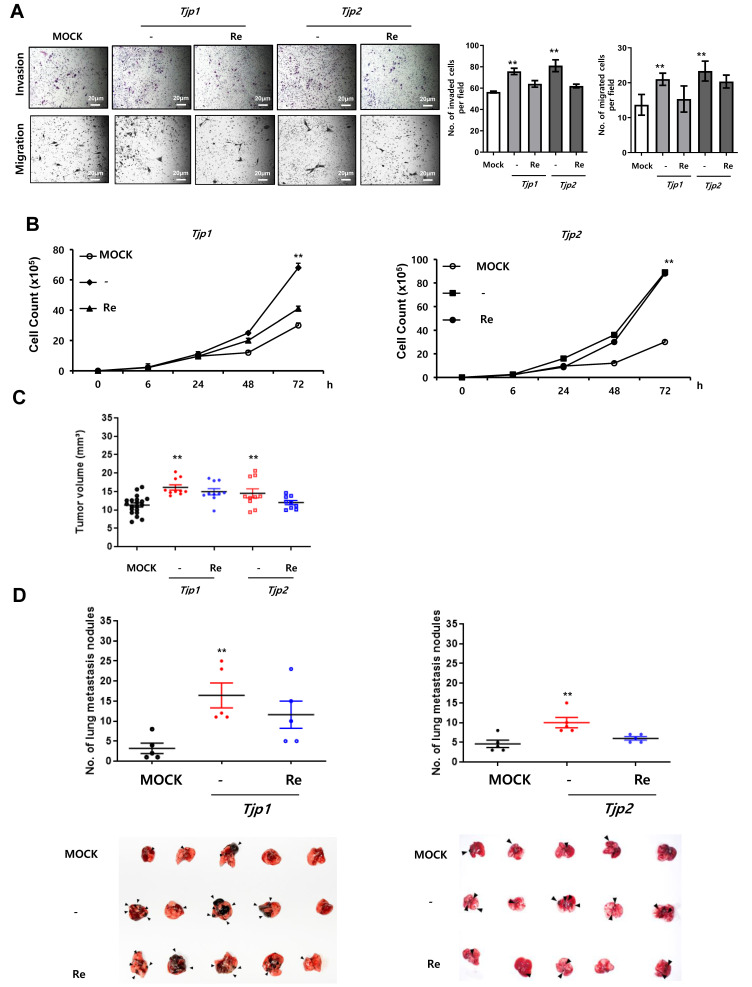
*Tjp1* and *Tjp2* KO increased tumorigenic characteristics, including invasion, migration, cell proliferation, tumor growth, and experimental metastasis, in B16-F10 cells. (**A**) Invasion and migration of MOCK, *Tjp1* KO, *Tjp2* KO, and re-expressed cells. Invasion and migration were significantly increased in *Tjp1* KO and *Tjp2* KO B16-F10 melanoma cells and were recovered in re-expressed cells of each KO group. (**B**) Cell proliferation of MOCK, *Tjp1* KO, *Tjp2* KO, and re-expressed cells. Cell proliferation was significantly increased in *Tjp1* KO and *Tjp2* KO cells and was recovered in re-expressed cells of each KO group. (**C**) Subcutaneous tumor growth assay. Dot plot of xenograft tumor volume of MOCK (black), *Tjp1* KO, *Tjp2* KO (red), and re-expressed (blue) cells at 14 days after subcutaneous injection of cells. Tumor size was significantly increased in *Tjp* KO cells and was recovered in re-expressed cells. (**D**) Experimental metastasis assay. MOCK (black), *Tjp* KO (red), and re-expressed (blue) cells were injected into the tail veins of mice (10 mice per group), and the mice were sacrificed 3 weeks after injection. The number of metastatic lung nodules was determined by directly counting the nodules on the lung. ** *p* < 0.01.

**Figure 3 ijms-25-00833-f003:**
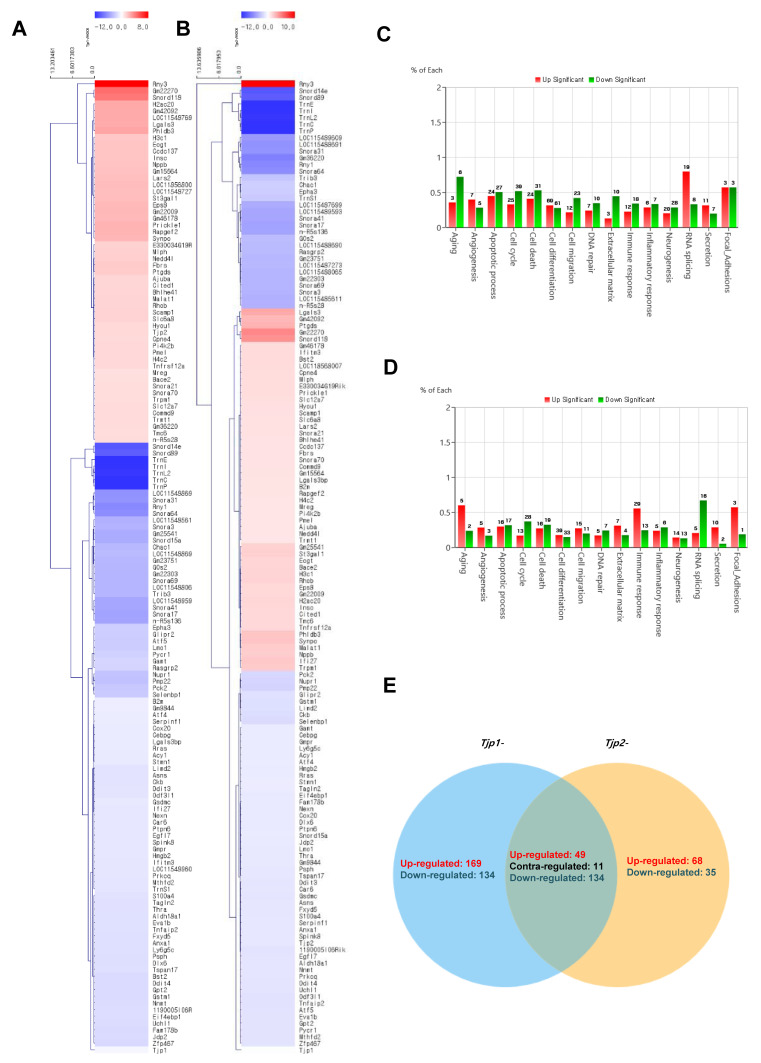
Transcriptome analysis of *Tjp*1 KO and *Tjp*2 KO B16-F10 melanoma cells. (**A**) Clustering analysis of differentially expressed mRNAs related to *Tjp1* KO. (**B**) Clustering analysis of differentially expressed mRNAs related to *Tjp2* KO. The fold change 2, log2-normalized read counts of four were selected in (**A**,**B**). (**C**) Gene ontology analysis of differentially expressed genes (DEGs) in *Tjp1* KO cells. (**D**) Gene ontology analysis of DEGs in *Tjp2* KO cells. (**E**) Comparison of the DEG expression patterns observed in the *Tjp1* KO and *Tjp2* KO cells.

**Figure 4 ijms-25-00833-f004:**
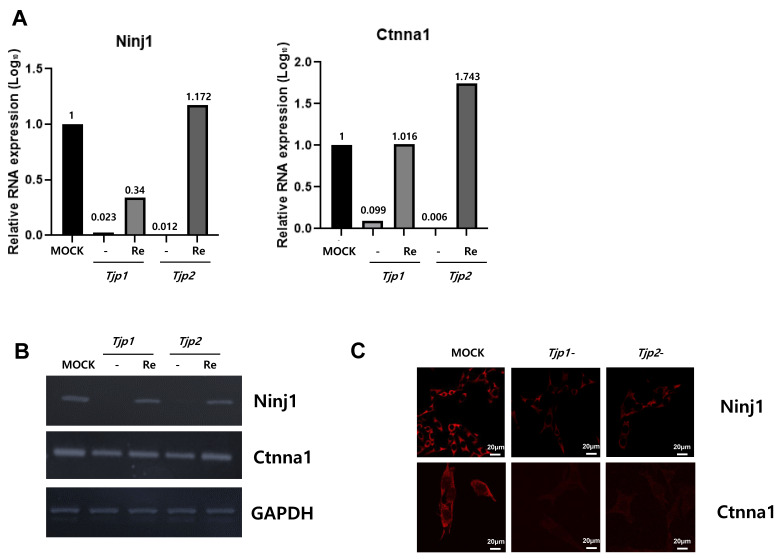
Expression of *Ninj1* and *Ctnna1* in *Tjp1* KO, *Tjp2* KO, and re-expressed B16-F10 melanoma cells. (**A**) Real-time PCR of *Ninj1* and *Ctnna1* in *Tjp1* KO, *Tjp2* KO, and re-expressed B16-F10 melanoma cells. The mRNA expression levels were calculated by normalizing to GAPDH and presented relative to the control with calculated mean values and 95% confidence intervals. (**B**) RT-PCR of *Ninj1* and *Ctnna1* in *Tjp1* KO, *Tjp2* KO, and re-expressed B16-F10 melanoma cells. (**C**) Immunofluorescence staining of Ninj1 and Ctnna1 in *Tjp1* and *Tjp2* KO cells.

**Figure 5 ijms-25-00833-f005:**
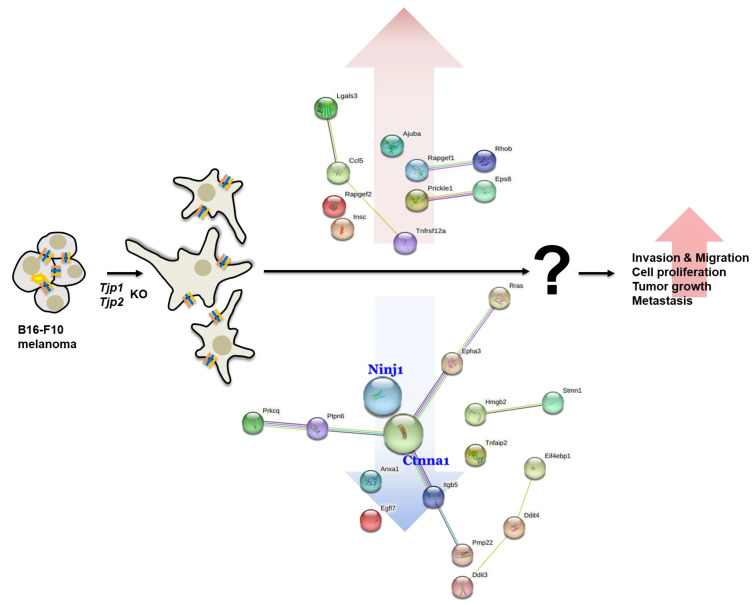
Schematic diagram of the effect of *Tjp1* and *Tjp2* KO on tumorigenic characteristics.

**Table 1 ijms-25-00833-t001:** Differential gene expression in *Tjp1* knockout cells.

Category	Gene	NCBI BLAST	Fold Change	Description	Classification
Cell cycle	*Eps8*	NM_001271588.1	4.672	Epidermal growth factor receptor pathway substrate 8	Up-regulation
	*Insc*	XM_006507662.5	3.545	INSC spindle orientation adaptor protein	Up-regulation
	*Rhob*	NM_007483.3	2.595	Ras homolog family member B	Up-regulation
	*Ajuba*	NM_010590.5	2.446	Ajuba LIM protein	Up-regulation
Cell migration	*Lgals3*	NM_001145953.1	6.333	Lectin, galactose binding, soluble 3	Up-regulation
	*Rapgef2*	XM_006502250.5	5.149	Rap guanine nucleotide exchange factor (GEF) 2	Up-regulation
	*Eps8*	NM_001271588.1	4.672	Epidermal growth factor receptor pathway substrate 8	Up-regulation
	*Rhob*	NM_007483.3	2.595	Ras homolog family member B	Up-regulation
	*Ajuba*	NM_010590.5	2.446	Ajuba LIM protein	Up-regulation
	*Tnfrsf12a*	NM_001161746.1	2.314	Tumor necrosis factor receptor superfamily, member 12a	Up-regulation
Angiogenesis	*Rhob*	NM_007483.3	2.595	Ras homolog family member B	Up-regulation
	*Tnfrsf12a*	NM_001161746.1	2.314	Tumor necrosis factor receptor superfamily, member 12a	Up-regulation
Cell–cell adhesion	*Ccl5*	NM_013653.3	1.646	Chemokine (C-C motif) ligand 5	Up-regulation
	*Rapgef1*	XM_030246487.1	1.85	Rap guanine nucleotide exchange factor (GEF) 1	Up-regulation
	*Prickle1*	NM_001364846.1	5.15	prickle planar cell polarity protein 1	Up-regulation
Cell cycle	*Stmn1*	NM_019641.4	0.441	Stathmin 1	Down-regulation
	*Ptpn6*	NM_001077705.2	0.389	Protein tyrosine phosphatase, non-receptor type 6	Down-regulation
	*Ddit3*	XM_006513197.4	0.342	DNA-damage inducible transcript 3	Down-regulation
	*Eif4ebp1*	NM_007918.3	0.258	Eukaryotic translation initiation factor 4E binding protein 1	Down-regulation
	*Pmp22*	NM_001302257.1	0.082	Peripheral myelin protein 22	Down-regulation
Cell migration	*Rras*	XM_036152813.1	0.443	Ras-related protein R-Ras isoform X1	Down-regulation
	*Egfl7*	NM_178444.5	0.386	Epidermal growth factor-like protein 7 isoform 1 precursor	Down-regulation
	*Hmgb2*	NM_001363443.1	0.379	High mobility group protein B2	Down-regulation
	*Prkcq*	XM_006497398.4	0.369	Protein kinase C, theta	Down-regulation
	*Ddit3*	XM_006513197.4	0.342	DNA-damage inducible transcript 3	Down-regulation
	*Anxa1*	NM_010730.2	0.28	Annexin A1	Down-regulation
	*Ddit4*	NM_029083.2	0.243	DNA-damage-inducible transcript 4	Down-regulation
	*Epha3*	NM_001362452.1	0.143	Eph receptor A3	Down-regulation
	*Pmp22*	NM_001302257.1	0.082	Peripheral myelin protein 22	Down-regulation
Angiogenesis	*Egfl7*	NM_178444.5	0.386	EGF-like domain 7	Down-regulation
	*Tnfaip2*	XM_006515793.5	0.298	Tumor necrosis factor, alpha-induced protein 2	Down-regulation
Cell–cell adhesion	*Ninj1*	NM_013610.3	0.487	Ninjurin 1	Down-regulation
	*Ctnna1*	XM_006525548.4	0.058	Catenin (cadherin associated protein), alpha 1,	Down-regulation
	*Itgb5*	NM_001145884.1	0.24	Integrin beta 5, transcript variant 1	Down-regulation
	*Ptpn6*	NM_001077705.2	0.389	Protein tyrosine phosphatase, non-receptor type 6	Down-regulation

**Table 2 ijms-25-00833-t002:** Differential gene expression in *Tjp2* knockout cells.

Category	Gene	NCBI BLAST	Fold Change	Description	Classification
Cell cycle	*Eps8*	NM_001271588.1	2.96	Epidermal growth factor receptor pathway substrate 8	Up-regulation
	*Insc*	XM_006507662.5	3.017	INSC spindle orientation adaptor protein	Up-regulation
	*Rhob*	NM_007483.3	3.283	Ras homolog family member B	Up-regulation
	*Ajuba*	NM_010590.5	2.074	Ajuba LIM protein	Up-regulation
Cell migration	*Lgals3*	NM_001145953.1	10.516	Lectin, galactose binding, soluble 3	Up-regulation
	*Rapgef2*	XM_006502250.5	2.095	Rap guanine nucleotide exchange factor (GEF) 2	Up-regulation
	*Eps8*	NM_001271588.1	2.96	Epidermal growth factor receptor pathway substrate 8	Up-regulation
	*Rhob*	NM_007483.3	3.283	Ras homolog family member B	Up-regulation
	*Ajuba*	NM_010590.5	2.074	Ajuba LIM protein	Up-regulation
	*Tnfrsf12a*	NM_001161746.1	2.82	Tumor necrosis factor receptor superfamily, member 12a	Up-regulation
Angiogenesis	*Rhob*	NM_007483.3	3.283	Ras homolog family member B	Up-regulation
	*Tnfrsf12a*	NM_001161746.1	2.82	Tumor necrosis factor receptor superfamily, member 12a	Up-regulation
Cell–cell adhesion	*Ccl5*	NM_013653.3	8.502	Chemokine (C-C motif) ligand 5	Up-regulation
	*Rapgef1*	XM_030246487.1	2.678	Rap guanine nucleotide exchange factor (GEF) 1	Up-regulation
	*Prickle1*	NM_001364846.1	2.476	Prickle planar cell polarity protein 1	Up-regulation
Cell cycle	*Stmn1*	NM_019641.4	0.492	Stathmin 1	Down-regulation
	*Ptpn6*	NM_001077705.2	0.428	Protein tyrosine phosphatase, non-receptor type 6	Down-regulation
	*Ddit3*	XM_006513197.4	0.36	DNA-damage inducible transcript 3	Down-regulation
	*Eif4ebp1*	NM_007918.3	0.437	Eukaryotic translation initiation factor 4E binding protein 1	Down-regulation
	*Pmp22*	NM_001302257.1	0.177	Peripheral myelin protein 22	Down-regulation
Cell migration	*Rras*	XM_036152813.1	0.472	Ras-related protein R-Ras isoform X1	Down-regulation
	*Egfl7*	NM_178444.5	0.341	Epidermal growth factor-like protein 7 isoform 1 precursor	Down-regulation
	*Hmgb2*	NM_001363443.1	0.471	High mobility group protein B2	Down-regulation
	*Prkcq*	XM_006497398.4	0.355	Protein kinase C, theta	Down-regulation
	*Ddit3*	XM_006513197.4	0.36	DNA-damage inducible transcript 3	Down-regulation
	*Anxa1*	NM_010730.2	0.389	Annexin A1	Down-regulation
	*Ddit4*	NM_029083.2	0.348	DNA-damage-inducible transcript 4	Down-regulation
	*Epha3*	NM_001362452.1	0.135	Eph receptor A3	Down-regulation
	*Pmp22*	NM_001302257.1	0.177	Peripheral myelin protein 22	Down-regulation
Angiogenesis	*Egfl7*	NM_178444.5	0.341	EGF-like domain 7	Down-regulation
	*Tnfaip2*	XM_006515793.5	0.315	Tumor necrosis factor, alpha-induced protein 2	Down-regulation
Cell–cell adhesion	*Ninj1*	NM_013610.3	0.653	Ninjurin 1	Down-regulation
	*Ctnna1*	XM_006525548.4	0.697	Catenin (cadherin associated protein), alpha 1,	Down-regulation
	*Itgb5*	NM_001145884.1	0.592	Integrin beta 5, transcript variant 1	Down-regulation
	*Ptpn6*	NM_001077705.2	0.389	Protein tyrosine phosphatase, non-receptor type 6	Down-regulation

## Data Availability

The data generated in this study are publicly available in Gene Expression Omnibus (GEO) at GSE240070.

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
