# Peer review of "Functional Analysis of Membrane-Associated Scaffolding Tight Junction (TJ) Proteins in Tumorigenic Characteristics of B16-F10 Mouse Melanoma Cells"

_ijms, 2024, doi:10.3390/ijms25020833_

Round 1

Reviewer 1 Report

Comments and Suggestions for Authors

Review of the manuscript in International Journal of Molecular Sciences, ijms-2755227

The manuscript entitled “Functional analysis of membrane-associated scaffolding tight junction (TJ) proteins in tumorigenic characteristics of B16-F10 3 mouse melanoma cells” provides the impact of tight junction-associated scaffold proteins on tumorigenic characteristics of B16-F10 melanoma cells. Overall, the results and data presentation are not clear and conflicting with the authors’ conclusion. 

Major points

1.     The manuscript should provide the painpoint, limitation, or reasons why this research needs to be investigated and how the data answered the main problems. In addition, the main point of this study needs to be demonstrated clearly.

2.     The resolution of all the figures in the PDF file is extremely unclear and difficult to see.

3.     Figure 1 showed the result of incomplete disruption in Tjp1 and Tjp2 knockout (KO). Tjp in KO strain was still expressed and Tjp in the re-expressing strain were not fully recovered (Fig. 1B and 1E). Tjp2 KO still weakly produced the protein (Fig. 1D). In addition, although immunofluorescence (IF) staining was recovered in the re-expressed cell line, the fluorescence was also shown in Tjp KO. These results are not convincing. Apparently, Tjp genes still have some functions in the KO strains; therefore, we cannot ensure that all results in this study are impacted by Tjp KO. 

4.     In Figs. 1F and 4C, were all the photo taken with exactly the same parameters? This is critical to compare among photos.

5.     Line 197, the author concluded that cell proliferation was significantly increased in Tjp1 KO and Tjp2KO cells and was recovered in the re-expressed cells of each KO group.  However, Figure 2B shows that cell proliferation in the re-expressed cell of Tjp2 was not recovered to mock, instead, it was similar to Tjp2 KO. Also, the lung metastasis nodules in re-expressed cells of Tjp1 were not recovered to mock (Fig. 2D). Therefore, these results were not clear that Tjp KO is involved in the stimulation of cell proliferation and lung metastasis. 

6.     The main point of transcriptome analysis should be provided. What is the important point of these genes in Table 1 and 2 that authors would like to demonstrate? How does the up-or down-regulation of these genes relate to TjpThe description and discussion in transcriptome analysis need to be more clearly explained. 

7.     Because melanoma cancer is an immunogenic cancer with a relationship to immune cells, the experiment on the impact of Tjp KO on immune cells such as CD4+ cell, CD8+ cell, NK cell, and cytokines (IL-2, TNF, IFN-g) should be demonstrated.     

8.     I strongly urge that the writing and language should be thoroughly edited by a native speaker. 

For example, “Table.1, 2 Details of the quantitatively different by Tjp1/Tip2 knockout”, are they awkward and grammatically incorrect?

Line 240, “Ratio control” -??  Is it “Gene expression ratio/fold change relative to control”?

9.     Line spacing in this manuscript is mostly strange, e.g., too tight between lines.

Comments on the Quality of English Language

The writing and language should be thoroughly edited by a native speaker. 

Author Response

Dear Reviewer 1,

  1. The manuscript should provide the painpoint, limitation, or reasons why this research needs to be investigated and how the data answered the main problems. In addition, the main point of this study needs to be demonstrated clearly.

: Tjp is a protein involved in cell-to-cell binding, and there is still a need for a thorough investigation into how cell-to-cell binding influences the characteristics of cancer growth, metastasis, migration, etc. in cancer cells.Therefore, our study affirms that the characteristics of cancer are enhanced by the knockout (KO) of Tjp1 and Tjp2 using CRISPR/Cas9. This information constitutes a new addition to the introduction section, specifically on Line 123.

  1. The resolution of all the figures in the PDF file is extremely unclear and difficult to see.

: We have attached a high-definition PDF file to the file that can be sent to the reviewer.

  1. Figure 1 showed the result of incomplete disruption in Tjp1and Tjp2 knockout (KO). Tjp in KO strain was still expressed and Tjp in the re-expressing strain were not fully recovered (Fig. 1B and 1E). Tjp2 KO still weakly produced the protein (Fig. 1D). In addition, although immunofluorescence (IF) staining was recovered in the re-expressed cell line, the fluorescence was also shown in Tjp KO. These results are not convincing. Apparently, Tjp genes still have some functions in the KO strains; therefore, we cannot ensure that all results in this study are impacted by Tjp KO. 

: The CRISPR/Cas9 knockout (KO) induces a double-strand break in a specific gene, resulting in the gene's removal through abnormal DNA repair. This process may lead to minimal gene expression, either through a heterogeneous recovery method of DNA repair or an alternative removal method for genes present in pairs. Additionally, trace amounts of gene expression may occur due to contamination during the clone selection process.

To address these potential variations, we systematically investigated dozens of clones, selecting the optimal one for testing. We conducted experiments, concluding that CRISPR-based knockout was more efficient than siRNA-based knockdown for functional investigations, even when a minimal amount of expression persisted.

In real-time PCR and Western blot analyses, it is inevitable to observe a very small amount of expression. However, we believe that this minimal expression is sufficient to discern the function of TJP in B16-F10 through the removal of TJP1 or TJP2. In cases of re-expression, it is unnecessary to fully restore the damaged gene; rather, a partial recovery aligning with conditions such as the transformation field suffices. The authors posit that observing whether the function is partially restored when the removed gene is re-expressed holds significance.

Regarding immunoluminescence, the presence of background signals due to antibody specificity or fluorescence non-specificity is a natural outcome. The fluorescence staining results of this study are considered insufficient to conclusively demonstrate significantly reduced expression and partial recovery through re-expression. These results are supplementary to those already proven by Western blot.

As mentioned earlier, this study substantially reduced TJP1 expression using CRISPR/Cas9, partially restoring it through re-expression. We consistently demonstrated changes in the cancer characteristics of B16-F10 cells based on these results. We acknowledge the reviewer's concern that not all results may be exclusively attributed to Tjp KO, but we are confident that our study provides valuable insights into the functional implications of TJP in B16-F10 cells.

Please raise the reviewer's claim that not all results of this study can be guaranteed to be affected by Tjp KO.

  1. In Figs. 1F and 4C, were all the photo taken with exactly the same parameters? This is critical to compare among photos.

: In both figures, the concentration of antibodies and experimental conditions were maintained consistently, as outlined in the materials and methods section. However, variations in conditions such as contrast or brightness during the photography process may be present.

  1. Line 197, the author concluded that cell proliferation was significantly increased in Tjp1KO and Tjp2KO cells and was recovered in the re-expressed cells of each KO group.  However, Figure 2B shows that cell proliferation in the re-expressed cell of Tjp2 was not recovered to mock, instead, it was similar to Tjp2 KO. Also, the lung metastasis nodules in re-expressed cells of Tjp1 were not recovered to mock (Fig. 2D). Therefore, these results were not clear that Tjp KO is involved in the stimulation of cell proliferation and lung metastasis. 

: As previously mentioned, re-expression does not entail a complete recovery of the gene removed by CRISPR/Cas9 but rather involves a partial restoration through transient transfection of an overexpression vector. Hence, it is expected that not all functions will return to the MOCK. However, the purpose of this experiment is to observe whether the increased cell proliferation resulting from knockout is partially reversed by the process of re-expression.

  1. The main point of transcriptome analysis should be provided. What is the important point of these genes in Table 1 and 2 that authors would like to demonstrate? How does the up-or down-regulation of these genes relate to Tjp? The description and discussion in transcriptome analysis need to be more clearly explained. 

: Our result table includes genes that commonly exhibit increases and decreases in tumorigenesis-related characteristics, such as cell cycle regulation, cell migration, angiogenesis, and cell-cell adhesion. This selection takes into account the cancer-related phenotypes manifested among the numerous genes expressed during Tjp knockout (tjp ko). Subsequently, the most crucial genes identified from these results underwent further experimental validation, and these additional findings have been incorporated into the results section. (line 233)

  1. Because melanoma cancer is an immunogenic cancer with a relationship to immune cells, the experiment on the impact of Tjp KO on immune cells such as CD4+ cell, CD8+ cell, NK cell, and cytokines (IL-2, TNF, IFN-g) should be demonstrated.     

: While it is intriguing to study the immune function in most cancers, including melanoma, this paper specifically delves into the examination of the growth and metastasis of cancer cells. The study focuses on TJP, which plays a role in cell-to-cell binding in cancer, and its relevance to migration and infiltration. Therefore, investigating immune phenomena may not be considered appropriate for the scope of this study.

  1. I strongly urge that the writing and language should be thoroughly edited by a native speaker. 

For example, “Table.1, 2 Details of the quantitatively different by Tjp1/Tip2 knockout”, are they awkward and grammatically incorrect?

Line 240, “Ratio control” -??  Is it “Gene expression ratio/fold change relative to control”?

: This paper underwent supervision by a professional native-speaking correction company and is accompanied by a correction certificate. However, for this specific section, further refinement was undertaken to enhance the overall fluency and clarity. And we changed Ratio control > Fold Change

  1. Line spacing in this manuscript is mostly strange, e.g., too tight between lines.

 : We confirmed.

Reviewer 2 Report

Comments and Suggestions for Authors

Study investigating the impact of CRISPR-Cas9-mediated knockout (KO) of Tjp1 and Tjp2 genes on tumorigenic characteristics in B16-F10 melanoma cells. The study analyzed various tumorigenic traits, including cell proliferation, migration, invasion, metastatic potential, and alterations in the expression patterns of associated proteins following the KO of Tjp genes. Transcriptome analysis via next-generation sequencing (NGS) was performed to elucidate the underlying molecular mechanisms affected by Tjp1 and Tjp2.

Methodology Assessment:

The methodology utilized CRISPR-Cas9-mediated KO of Tjp1 and Tjp2 genes in B16-F10 melanoma cells, followed by a comprehensive analysis of tumorigenic characteristics in vitro and in vivo. The study employed next-generation sequencing (NGS) for transcriptome analysis to uncover key genes affected by Tjp KO. Overall, the methodology seems robust and well-designed, offering a suitable framework to investigate the functional roles of Tjp1 and Tjp2 in melanoma tumorigenicity.

Findings Summary:

The findings revealed noteworthy alterations upon Tjp1 and Tjp2 KO, showcasing a significant increase in cell proliferation, migration, invasion, tumor growth, and metastatic potential in the KO cells. The decrease in Zonula occludens (ZO)-associated protein Claudin-1 (CLDN-1) in Tjp KO cells suggests a potential mechanism underlying altered cell-to-cell adhesion. Notably, transcriptome analysis identified significant changes in genes related to cell cycle regulation, cell migration, angiogenesis, and cell-cell adhesion, with prominent downregulation of Ninjurin-1 (Ninj1) and Catenin alpha-1 (Ctnna1) genes upon Tjp KO.

Strengths:

  1. Comprehensive Analysis: The study covers various tumorigenic characteristics, providing a holistic view of the impact of Tjp1 and Tjp2 KO.
  2. Molecular Insights: Transcriptome analysis sheds light on the molecular pathways influenced by Tjp1 and Tjp2, highlighting key genes involved.
  3. In Vivo Relevance: The findings from in vivo mouse models strengthen the relevance of observed alterations in tumorigenicity.

Limitations:

  1. Mechanistic Details: While the study identifies altered genes and tumorigenic traits, it could benefit from deeper mechanistic elucidation of how Tjp1 and Tjp2 precisely regulate these processes.
  2. Clinical Implications: Further exploration is required to assess the potential translational implications of these findings in clinical melanoma treatments.

This study presents valuable insights into the roles of Tjp1 and Tjp2 in regulating tumorigenic characteristics in B16-F10 melanoma cells. The findings significantly contribute to understanding the molecular underpinnings of melanoma progression. Further investigations addressing the specific mechanisms and potential therapeutic implications are warranted to harness these findings for clinical applications in melanoma treatment.

Author Response

Dear Reviewer 2,

Study investigating the impact of CRISPR-Cas9-mediated knockout (KO) of Tjp1 and Tjp2 genes on tumorigenic characteristics in B16-F10 melanoma cells. The study analyzed various tumorigenic traits, including cell proliferation, migration, invasion, metastatic potential, and alterations in the expression patterns of associated proteins following the KO of Tjp genes. Transcriptome analysis via next-generation sequencing (NGS) was performed to elucidate the underlying molecular mechanisms affected by Tjp1 and Tjp2.

Methodology Assessment:

The methodology utilized CRISPR-Cas9-mediated KO of Tjp1 and Tjp2 genes in B16-F10 melanoma cells, followed by a comprehensive analysis of tumorigenic characteristics in vitro and in vivo. The study employed next-generation sequencing (NGS) for transcriptome analysis to uncover key genes affected by Tjp KO. Overall, the methodology seems robust and well-designed, offering a suitable framework to investigate the functional roles of Tjp1 and Tjp2 in melanoma tumorigenicity.

Findings Summary:

The findings revealed noteworthy alterations upon Tjp1 and Tjp2 KO, showcasing a significant increase in cell proliferation, migration, invasion, tumor growth, and metastatic potential in the KO cells. The decrease in Zonula occludens (ZO)-associated protein Claudin-1 (CLDN-1) in Tjp KO cells suggests a potential mechanism underlying altered cell-to-cell adhesion. Notably, transcriptome analysis identified significant changes in genes related to cell cycle regulation, cell migration, angiogenesis, and cell-cell adhesion, with prominent downregulation of Ninjurin-1 (Ninj1) and Catenin alpha-1 (Ctnna1) genes upon Tjp KO.

Strengths:

  1. Comprehensive Analysis: The study covers various tumorigenic characteristics, providing a holistic view of the impact of Tjp1 and Tjp2 KO.
  2. Molecular Insights: Transcriptome analysis sheds light on the molecular pathways influenced by Tjp1 and Tjp2, highlighting key genes involved.
  3. In Vivo Relevance: The findings from in vivo mouse models strengthen the relevance of observed alterations in tumorigenicity.

Limitations:

  1. Mechanistic Details: While the study identifies altered genes and tumorigenic traits, it could benefit from deeper mechanistic elucidation of how Tjp1 and Tjp2 precisely regulate these processes.
  2. Clinical Implications: Further exploration is required to assess the potential translational implications of these findings in clinical melanoma treatments.

: Thank you for your comment. We added related contents to the reference reflection the reviewer’s suggestion. (line 351~)

Reviewer 3 Report

Comments and Suggestions for Authors

The manuscript by Ko et al titled “Functional analysis of membrane-associated scaffolding tight junction (TJ) proteins in tumorigenic characteristics of B16-F10 mouse melanoma cells” presents a study on the role of ZO-1 and ZO-2 proteins on tumorigenic characteristics of B16-F10 cells. By knocking out TJP1 or TJP2 genes encoding ZO-1 and ZO-2 proteins in melanoma cells and characterizing the KO cells, they conclude that ZO-1 and ZO-2 decrease the cell proliferation and metastatic potential of B16-F10 cells. They also observed that the KO cells have decreased expression of claudin-1, which could not be recovered after re-expression.

The studies provided in this manuscript of general interest. However, the quality of the paper could be significantly improved after clarifying the following issues and addressing the comments.

1-    The figures have an extremely low quality to the extent that they are not readable. For example, the header of Figure 1 is not decipherable by any means even after zooming in on the pdf. The axes of the plots in Figure 1B, 1D, 1E, and 1G are barely readable. The subfigures should be realigned and arranged in a better way, probably horizontally, to make it easier to follow.

Figure 2, impossible to read or even compare the graphics in panels A and D. Again, the subplots are not even aligned. 

Figure 3A is not usable in the current format. Nothing can be read on the figure, too small in the current format and quality extremely low.

2-    The authors start the manuscript with the following statement: “Tight junction proteins (Tjps), also known as zonula occludens (ZO) are scaffolding proteins that are involved in the formation and maintenance of tight junctions”. My understanding is that the authors refer to TJP1 and TJP2 genes encoding ZO-1 and ZO-2. However, their use of Tjps for tight junction scaffolding proteins is inaccurate and leads to confusion, since in the literature “TJ proteins” refer to all proteins in the TJ complex and not just the scaffolding ZO proteins, this includes its membrane proteins as well as cytoplasmic scaffolding proteins.

They have also used various notations for tight junction proteins, sometimes using two different notations in the same sentence: Tjps, TJP, TJ protein, TJ molecule, etc. 

3-    Following on the previous comment, the first paragraph of the paper is also written as if it is an overview of the TJ proteins – including those involved in channel formation, i.e. claudins, etc. It is inconsistent in multiple instances, e.g. after introducing Tjp1, Tjp2, and Tjp3, the authors state that “Tjps are involved in the maintenance of tight-junction integrity, regulation of paracellular transport, and control of epithelial permeability.”

Scaffolding proteins in TJs do not regulate paracellular transport and do not control epithelial permeability. Any such effect would be indirect by affecting the TJ organization, and it must be clarified in this paragraph. Proper citation of previous work addressing TJ integrity or organization is also needed, e.g. no references are provided for the the following two statements at the end of this paragraph: “Previous studies ... have focused on the regulation of barrier function. Tjps are involved in the maintenance of tight junction integrity ....” 

4-    It would be helpful to provide a little more information of the structure of ZO proteins, e.g. on the fact that their N-terminal contains PDZ domains that facilitate protein-protein association, etc. 

5-    It is stated (p2) that “These proteins also include those involved in epithelial proliferation and differentiation, and Tjps are known to function beyond regulating cell permeability [10]”. ZO proteins are not involved in permeability, and most importantly not in cell permeability (TJs control paracellular permeability, i.e. transport between the cells). This statement is not accurate and has to be revised. Again, too much emphasis on permeability is misleading and not necessary, since it is only due to the indirect effect of ZO proteins on TJ overall structure. 

6-    With no prior discussion or relevance, the authors mention (p2) that “In addition, by transmitting information on the degree of cell-cell contact with nucleus, Tjps maintain a balance between proliferation and differentiation [8].” Please elaborate on this (connection to nucleus) or remove this statement. 

7-    Near the end of p2, the authors mention the possible role of Tjp1 in cancer by “disruption of cell-cell adhesion and increased cell migration, invasion and metastasis”, which are all very reasonable suggestions. However, they never discuss the role of TJs in defining the cell polarity and disruption of that can contribute to cancer cell proliferation or metastasis. This could be a place to discuss that. 

8-    On p3, “TJ proteins”, “tight junction molecules”, and Tjps are all used interchangeably in the same paragraph and truly leads to confusion.

9-    The authors show that the knock on TJP1 and TJP2 cells have a decreased level of claudin-1. They also show re-expression of TJP1 and TJP2 does not recover claudin-1. This is contradictory to some previously published work, e.g. the work of Umeda et al, 2006, in which after deleting ZO-1 and ZO-2 from mouse mammary gland tumor Eph4 cells, claudin expression was normal. In fact, Umeda et al distinguish between claudin expression level and TJ formation and conclude that deleting ZO proteins results in no tight junction despite normal claudin levels. 

This is worth a discussion in the paper and comparison with previous results. Most importantly, it is very plausible to understand how deleting ZO proteins can disrupt TJ formation, but it is not clear how it can affect claudin expression. Current understanding in the field is that ZO proteins are mainly involved in recruiting claudins to the TJ. If this is a novel result, one expects the authors to speculate on possible role of ZO in regulating claudin expression and providing the context on why it contradicts previously published work. 

10- Minor comment: the authors in a few occasions mentioned that “the tumor growth was significantly increased in the Tjps KO injected groups and recovered in the Tjps re-expressed groups.” I found it confusing to refer to a slowdown of the growth rate as recovery. It would have been more accurate to say that the growth rates were recovered or that growth [rate] was slowed down.

Author Response

Dear Reviewer 3,

1-    The figures have an extremely low quality to the extent that they are not readable. For example, the header of Figure 1 is not decipherable by any means even after zooming in on the pdf. The axes of the plots in Figure 1B, 1D, 1E, and 1G are barely readable. The subfigures should be realigned and arranged in a better way, probably horizontally, to make it easier to follow.

Figure 2, impossible to read or even compare the graphics in panels A and D. Again, the subplots are not even aligned. 

Figure 3A is not usable in the current format. Nothing can be read on the figure, too small in the current format and quality extremely low.

: We are uncertain about the reason behind the reduced readability of the high-resolution picture during the posting process. To ensure clarity, high-resolution images have been attached to facilitate easy reading of all pictures. For Figure 3A, the resolution of the heatmap, provided by the program, cannot be further increased. Consequently, not only have we compiled promising genes in the table, but we have also submitted the relevant data to NCBI GEO to enable the retrieval of information on all genes.

2-    The authors start the manuscript with the following statement: “Tight junction proteins (Tjps), also known as zonula occludens (ZO) are scaffolding proteins that are involved in the formation and maintenance of tight junctions”. My understanding is that the authors refer to TJP1 and TJP2 genes encoding ZO-1 and ZO-2. However, their use of Tjps for tight junction scaffolding proteins is inaccurate and leads to confusion, since in the literature “TJ proteins” refer to all proteins in the TJ complex and not just the scaffolding ZO proteins, this includes its membrane proteins as well as cytoplasmic scaffolding proteins.

They have also used various notations for tight junction proteins, sometimes using two different notations in the same sentence: Tjps, TJP, TJ protein, TJ molecule, etc. 

: We appreciate the reviewer's concerns. While using a uniform notation like 'zonula occludens (ZO)' might prevent any confusion, it's important to note that the human ZO gene is officially designated as TJP in the NCBI gene and mouse ZO gene is officially designated as Tjp. The authors opted for commonly used gene names across various papers to maintain consistency. Unfortunately, adopting names not utilized in other publications might lead to confusion with other proteins involved in tight junctions. Nonetheless, we have made modifications to the notation to present it as a single TJP gene or protein wherever possible.

3-    Following on the previous comment, the first paragraph of the paper is also written as if it is an overview of the TJ proteins – including those involved in channel formation, i.e. claudins, etc. It is inconsistent in multiple instances, e.g. after introducing Tjp1, Tjp2, and Tjp3, the authors state that “Tjps are involved in the maintenance of tight-junction integrity, regulation of paracellular transport, and control of epithelial permeability.”

Scaffolding proteins in TJs do not regulate paracellular transport and do not control epithelial permeability. Any such effect would be indirect by affecting the TJ organization, and it must be clarified in this paragraph. Proper citation of previous work addressing TJ integrity or organization is also needed, e.g. no references are provided for the the following two statements at the end of this paragraph: “Previous studies ... have focused on the regulation of barrier function. Tjps are involved in the maintenance of tight junction integrity ....” 

: To avoid confusion and clearly distinguish between TJP, encompassing both TJP and Claudin genes, and the abbreviations TJP1 or TJP2, all instances of the general term 'TJP' were modified to specify TJP1 and TJP2. Furthermore, to enhance clarity, the term 'tight junction protein' was utilized in full rather than its abbreviation [TJP].

4-    It would be helpful to provide a little more information of the structure of ZO proteins, e.g. on the fact that their N-terminal contains PDZ domains that facilitate protein-protein association, etc. 

: We added to the introduction sections (Line 54) 

5-    It is stated (p2) that “These proteins also include those involved in epithelial proliferation and differentiation, and Tjps are known to function beyond regulating cell permeability [10]”. ZO proteins are not involved in permeability, and most importantly not in cell permeability (TJs control paracellular permeability, i.e. transport between the cells). This statement is not accurate and has to be revised. Again, too much emphasis on permeability is misleading and not necessary, since it is only due to the indirect effect of ZO proteins on TJ overall structure. 

: The abbreviation [TJP] was not used in the manner as described above.

6-    With no prior discussion or relevance, the authors mention (p2) that “In addition, by transmitting information on the degree of cell-cell contact with nucleus, Tjps maintain a balance between proliferation and differentiation [8].” Please elaborate on this (connection to nucleus) or remove this statement. 

: We deleted.

7-    Near the end of p2, the authors mention the possible role of Tjp1 in cancer by “disruption of cell-cell adhesion and increased cell migration, invasion and metastasis”, which are all very reasonable suggestions. However, they never discuss the role of TJs in defining the cell polarity and disruption of that can contribute to cancer cell proliferation or metastasis. This could be a place to discuss that. 

: We added metastasis reference in discussion sections (line 315~) and we have also written our additional comments (line 351~)

8-    On p3, “TJ proteins”, “tight junction molecules”, and Tjps are all used interchangeably in the same paragraph and truly leads to confusion.

: We confirmed.  

9-    The authors show that the knock on TJP1 and TJP2 cells have a decreased level of claudin-1. They also show re-expression of TJP1 and TJP2 does not recover claudin-1. This is contradictory to some previously published work, e.g. the work of Umeda et al, 2006, in which after deleting ZO-1 and ZO-2 from mouse mammary gland tumor Eph4 cells, claudin expression was normal. In fact, Umeda et al distinguish between claudin expression level and TJ formation and conclude that deleting ZO proteins results in no tight junction despite normal claudin levels. 

This is worth a discussion in the paper and comparison with previous results. Most importantly, it is very plausible to understand how deleting ZO proteins can disrupt TJ formation, but it is not clear how it can affect claudin expression. Current understanding in the field is that ZO proteins are mainly involved in recruiting claudins to the TJ. If this is a novel result, one expects the authors to speculate on possible role of ZO in regulating claudin expression and providing the context on why it contradicts previously published work. 

: Alterations in the expression of other tight junction proteins due to ZO knockout are manifesting differently in various cells, as noted by the reviewer. Follow-up studies are deemed necessary. In response to this concern, I have included a reference.

In this study, we observe a reduction in claudin levels in the case of B16-F10 melanoma. Additionally, the re-expression of TJP1 and the failure of TJP2 to restore claudin-1 raise questions about whether the re-expression is inadequate for partial recovery of claudin-1 or if it falls short of fully restoring TJP1 and TJP2. Moreover, it remains uncertain whether the knockout-induced reduction of claudin-1 in TJP1 and TJP2 occurs through alternative pathways. These aspects necessitate further investigation, considering potential cell-specific variations, which we have added in the discussion sections (line 274~)

10- Minor comment: the authors in a few occasions mentioned that “the tumor growth was significantly increased in the Tjps KO injected groups and recovered in the Tjps re-expressed groups.” I found it confusing to refer to a slowdown of the growth rate as recovery. It would have been more accurate to say that the growth rates were recovered or that growth [rate] was slowed down.

: We changed, growth rate and metastasis rate.

Round 2

Reviewer 1 Report

Comments and Suggestions for Authors

Review of the manuscript in International Journal of Molecular Sciences, ijms-2755227_V2

The revised manuscript entitled “Functional analysis of membrane-associated scaffolding tight junction (TJ) proteins in tumorigenic characteristics of B16-F10 3 mouse melanoma cells” has been submitted to the journal in a relatively short time.  The revision included only the revised statement in the text. 

Major points

1.     The response regarding the data in Fig. 1, particularly 1E and 1F, was not clear. The gene expression and immuno-fluorescence in the Tjp1-related strains, KO vs complemented, were not different (1E and 1F).  Likewise, the immuno-fluorescence in the Tjp2-related strains, KO vs complemented, were not different (1F).  These results affect the authors’ conclusion in the role of these two genes.

When you do the complementation experiment right, the affected phenotypes should be restored to the WT level.  If not, there could be something wrong with the complementing gene sequence or other mutations appear in the KO strains.

2.     The authors’ response on the point on the fluorescence microscopy “However, variations in conditions such as contrast or brightness during the photography process may be present.”  For those who are doing a lot of fluorescence and confocal microscopy, we know that these fluorescence photos need to be taken with high carefulness. They need standardization across several photographs we are taking, otherwise the data could provide very little meaning. So, if you do adjust the parameter settings between photos either during the microscopy or in the post-process, that really affects the interpretation of results. 

Comments on the Quality of English Language

Moderate editing is still required.

Author Response

  1. The response regarding the data in Fig. 1, particularly 1E and 1F, was not clear. The gene expression and immuno-fluorescence in the Tjp1-related strains, KO vs complemented, were not different (1E and 1F). Likewise, the immuno-fluorescence in the Tjp2-related strains, KO vs complemented, were not different (1F).  These results affect the authors’ conclusion in the role of these two genes.

When you do the complementation experiment right, the affected phenotypes should be restored to the WT level.  If not, there could be something wrong with the complementing gene sequence or other mutations appear in the KO strains.

The opinions of reviewer who believes there is no difference between KO and complemented are subjective, and we cannot agree. To substantiate this assertion, we also employed an image analysis program and verified distinct differences in q RT-PCR and immuno-fluorescence. As we answered previously, when re-expression experiments are conducted using the over-expression vector in knockout (KO) cells, it is challenging to achieve expression values similar to wild type. We’ve already answered, it is worthwhile to note that observing and experimenting with the phenomenon may reveal a restoration to WT-like levels when a specific threshold of expression is regained. We do not fully agree with the reviewer's assertion that we must achieve a statistically significant p-value and confirm expression restoration to WT values.

When examining various papers on Re-expression, one may observe instances where Re-expression has not necessarily led to recovery, as seen in the case of WT [Koyama S, Tsuchiya H, Amisaki M, Sakaguchi H, Honjo S, Fujiwara Y, Shiota G. NEAT1 is Required for the Expression of the Liver Cancer Stem Cell Marker CD44. Int J Mol Sci. 2020 Mar 11;21(6):1927. doi: 10.3390/ijms21061927. PMID: 32168951; PMCID: PMC7139689.] and [Al-Zeheimi N, Gao Y, Greer PA, Adham SA. Neuropilin-1 Knockout and Rescue Confirms Its Role to Promote Metastasis in MDA-MB-231 Breast Cancer Cells. Int J Mol Sci. 2023 Apr 25;24(9):7792. doi: 10.3390/ijms24097792. PMID: 37175499; PMCID: PMC10178772.]

In this experiment, it can be confirmed that the phenotype that occurs when knock-out (KO) is reintroduced through re-expression is recovered to some extent. Therefore, I believe there is no issue with the re-expression experiment.

About Fig 1E, we conducted qPCR experiments more than three times and generated the graph using mean values. Our analysis revealed a significant p-value. Fig 1F also confirmed by three co-authors by measuring the value using ImageJ and calculated the average expression value.

Density mean ;

Tjp1 group (MOCK 66, KO 40, Re 52)

Tjp2 group (MOCK 66, KO 29, Re 35)

  1. The authors’ response on the point on the fluorescence microscopy “However, variations in conditions such as contrast or brightness during the photography process may be present.” For those who are doing a lot of fluorescence and confocal microscopy, we know that these fluorescence photos need to be taken with high carefulness. They need standardization across several photographs we are taking, otherwise the data could provide very little meaning. So, if you do adjust the parameter settings between photos either during the microscopy or in the post-process, that really affects the interpretation of results.

We completely agree that the result value changes by parameters and calibration. For data fairness, we maintained consistent parameter setting across microscope captures and in post-processing.

This figure was also not calibrated to the original image at the time of confocal imaging.

We have submitted this paper to a professional English correction service and have provided evidence of the corrections made. Additionally, other reviewers have affirmed that there are no issues with the English language. Specifically, the English correction service we engaged is specialized in professionally correcting scientific papers. It is noteworthy that some of its editors are native speakers, ensuring a more specialized and objective correction of English. In light of these objective considerations, it is challenging to concur with the reviewer's suggestion that further English correction is necessary.